# The Use of Percutaneous Thermal Sensing Microchips for Body Temperature Measurements in Horses Prior to, during and after Treadmill Exercise

**DOI:** 10.3390/ani10122274

**Published:** 2020-12-02

**Authors:** Hyungsuk Kang, Rebeka R. Zsoldos, Solomon M. Woldeyohannes, John B. Gaughan, Albert Sole Guitart

**Affiliations:** 1School of Agriculture and Food Sciences, The University of Queensland, Gatton, QLD 4343, Australia; hyungsuk.kang@uq.net.au (H.K.); r.zsoldos@uq.edu.au (R.R.Z.); j.gaughan@uq.edu.au (J.B.G.); 2School of Veterinary Science, The University of Queensland, Gatton, QLD 4343, Australia; s.woldeyohannes@uq.edu.au

**Keywords:** horse, body temperature, percutaneous thermal sensing microchip, central venous temperature, treadmill exercise

## Abstract

**Simple Summary:**

During exercise, horses produce heat from working muscles, and this heat, under certain circumstances, may accumulate in the body. Conduction, convection, radiation, and evaporation are the primary heat transfer mechanisms for the control of body temperature. Horses that undergo strenuous exercise in hot and humid environments may have heat production that exceeds their ability to dissipate the heat. Therefore, the horse could be at risk from postexercise exertional heat illness, possibly leading to heat shock and death. To avoid this outcome, many Thoroughbred racehorses are cooled-down postracing by using water application (ice-cold water or ambient temperature water), fans, combinations of water application and fans, or water application followed by scraping the water off the horse. Early detection of the clinical signs of exertional heat illness and adequate treatments are important to prevent severe hyperthermia and irreversible thermal damage. The development and application of technology that will provide accurate, rapid, safe, and noninvasive monitoring of body temperature changes might help the detection of postrace exertional heat illness in equine athletes. Implanted percutaneous thermal sensing microchip (PTSM) is a reliable method to measure body temperature in horses; however, the optimal location within the body of the horse needs to be determined.

**Abstract:**

Accurately measuring body temperature in horses will improve the management of horses suffering from or being at risk of developing postrace exertional heat illness. PTSM has the potential for measuring body temperature accurately, safely, rapidly, and noninvasively. This study was undertaken to investigate the relation between the core body temperature and PTSM temperatures prior to, during, and immediately after exercise. The microchips were implanted into the nuchal ligament, the right splenius, gluteal, and pectoral muscles, and these locations were then compared with the central venous temperature, which is considered to be the “gold standard” for assessing core body temperature. The changes in temperature of each implant in the horses were evaluated in each phase (prior to, during, and immediately postexercise) and combining all phases. There were strong positive correlations ranging from 0.82 to 0.94 (*p* < 0.001) of all the muscle sites with the central venous temperature when combining all the phases. Additionally, during the whole period, PTSM had narrow limits of agreement (LOA) with central venous temperature, which inferred that PTSM is essentially equivalent in measuring horse body temperature. Overall, the pectoral PTSM provided a valid estimation of the core body temperature.

## 1. Introduction

Thoroughbred horse racing is one of the most attractive events among horse competitions in Australia [1,2]. According to Racing Australia [1,2], 19,409 and 19,369 races were held in Australia during the 2017/18 and 2018/19 season, respectively, and 35,107 and 35,196 horses were raced during the respective seasons. During the two seasons, horses were raced on average over a 1000 m track, which they completed in 56 s, equating to a speed of 17.86 m/s [1]. The horses raced under various weather conditions, including heatwaves. Several states in Australia have policies in place to guide the Thoroughbred racing industry to protect the welfare of the horses racing and training in hot weather [3,4,5,6,7].

It was reported by Hodges et al. [8] that strenuous exercise, such as Thoroughbred racing, requires horses to expend a large amount of energy, which produces metabolic heat in muscles leading to an increase in body temperature of 1 °C for every one minute of racing. Horses are normally able to control this accumulated body heat by dissipating heat to the environment via conduction, convection, radiation, and evaporation [8]. However, if the exercise is prolonged, especially during hot or hot and humid weather, horses may have limited capacity to dissipate the accumulated body heat [9]. Furthermore, unlike humans or other animal athletes, horses have a large body mass with a small body surface area, which further restricts the dissipation of body heat [10,11].

As stated, when exercise is prolonged during hot or hot and humid weather, heat dissipation may be limited and, eventually, heat transfer may be reversed, i.e., the animal does not dissipate heat to the environment and may actually gain heat from the environment [9,12,13]. Even if the transfer by convection and radiation are reversed, the horse may still be able to dissipate body heat via sweating even when weather conditions are such that the rate of sweat evaporation is reduced [13]. However, horses have thick, waterproof hair, which interrupts the evaporative cooling of sweat by blocking its exposure to air [14]. The heat dissipation efficiency of the sweat running off the skin is only 5 to 10% of the sweat evaporation [13,15], so evaporative heat loss is critical if horses are to maintain body temperature within a tolerable range. To support evaporation from the body surface during the hot or hot and humid condition, horses produce a surfactant protein, Latherin, and when mixed with sweat, it helps the wetting of the hair by sweat, improves evaporation, and decreases the amount of sweat running off the surface of the animal [8,14,16]. Under hot or hot and humid conditions, prolonged exercise may cause exertional heat illness (EHI), dehydration, collapse, and death [17,18]. When the central blood, the hypothalamic, and muscle temperatures reach 42.5 °C, 41.5 °C, and 45 °C, respectively, horses may show signs of hyperthermia [15]. It has been demonstrated that blood temperature is a very sensitive measure of core body temperature in horses during exercise [9,19,20,21,22].

Monitoring weather conditions before and during a race event is essential for the safety of the animal. In terms of exertional heat illness or heat-related regulations, the wet-bulb globe temperature (WBGT) is used worldwide for monitoring sports events, such as the International Olympic Committee or Fédération Equestre Internationale competitions. Some racing organizations in Australia also use the WBGT index to aid in the decision-making process of weather safety prior to a race. When the forecast WBGT is above 26 °C, it is recommended that organizers consider modifying or canceling the race. Governing bodies in Australia (Racing Queensland, Thoroughbred Racing South Australia, Racing New South Wales, and Racing Victoria) recommend taking extra care, such as providing an additional veterinarian at the track, ensuring the availability of adequate drinking water, and washing bays for horses at the events. In 2017, Equestrian Australia (EA) announced that when WBGT is above 28 °C, extra care and precautions are required to limit overheating of horses. However, there is an overall lack of guidelines for the recognition, prevention, and treatment of postrace EHI. As reported by Brownlow et al. [23], a horse with EHI shows increased rates of respiration and heart rate, slow capillary refill times, loss of consciousness, and some dangerous behaviors, such as kicking out randomly or lunging forward. To prevent brain damage, blood flow to the brain is restricted during hyperthermia, which can cause cerebral ischemia and brain edema. The induced brain damage results in central nervous system dysfunction leading to a loss of normal reflex responses, headache, collapse, or coma. Several methods can be used to reduce accumulated body heat after racing, such as taking the horses into an air-conditioned room, spraying cool water and fanning, pouring cool water on the skin, placing on ice collar of the neck, or using intravenous detomidine hydrochloride (a sedative) to reduce and to attenuate clinical signs of central nervous system (CNS) dysfunction [24,25,26,27,28,29]. However, it is not clear when more aggressive actions should be undertaken to save a horse from the effects of excessive heat load. Furthermore, there are currently no methods to detect early or mild postrace EHI cases. If body temperature could be measured accurately, safely, quickly, and noninvasively, then early detection and management of EHI would be possible.

Many methods are used to measure body temperature, such as gastrointestinal pills (gastrointestinal temperature), infrared thermal image (eye or skin temperature), central venous temperature, digital thermometer (rectal temperature), or using percutaneous thermal sensing microchips (PTSM) (muscle temperature) [22,30,31,32,33]. However, rectal thermometry is the most commonly used method to estimate core body temperature [34,35].

Rectal thermometry is an inexpensive technique, yet it is also time- and labor-consuming to perform, and may inflict local injury in horses, particularly during repeated samplings, and commonly induces handling stress or requires the use of sedatives. However, obtaining the rectal temperature of a horse immediately postexercise can also be dangerous and infeasible for the operator due to the restless behavior, especially in highly excitable horses and the constant movement of the animal while it is recovering from strenuous exercise. Furthermore, if the horse is exhibiting signs of postrace EHI, such as irritability and uncooperative behavior or kicking out, it will be unsafe and challenging to obtain the rectal temperature. Infrared temperature image has been used as a method to measure horse body temperature, but it may be affected by sunlight and therefore may be of limited practical value as an assessment of core body temperature [36].

In this study, it was hypothesized that PTSMs would accurately measure body temperature changes in horses. There are a lack of data on testing PTSMs usage in horses during exercise and immediately after exercise. In addition, no previous studies have tested PTSMs implanted in various locations of the horses’ body [35,37]. The aim of the present study was to document the relationship between the central venous temperature (as the core body temperature) and PTSMs temperatures in various muscle sites during and immediately after exercise.

## 2. Materials and Methods

### 2.1. Animals

Eight unconditioned adult horses (7 geldings and 1 mare; 4 Thoroughbred and 4 Standardbred) ranging from 4 to 12 years old (average 7 ± 3 years old) and 455 to 545 kg body weight (average 493 ± 31 kg) were used in the study. The horses are part of the research herd at The University of Queensland (Gatton Campus), and they are routinely housed in groups of 10 to 15 horses in large grazing paddocks located on the Gatton campus. Ad libitum access to pasture and hay is provided with oat hay (1.5% of body weight) once or twice a week. The horses used in the study were not exercised during the 3 months prior to the commencement of the study. During the experiment, the animals were housed in small yards adjacent to the UQ VETS, Equine Specialist Hospital, The University of Queensland Gatton campus. Water was available ad libitum, and 1.5% of body weight of lucerne hay was provided per day. The use of animals and all experimental procedures were approved by the Animal Ethics Committees (AEC) of The University of Queensland (Approval No. SAFS/431/18). All horses were shod prior to the commencement of the study. A general physical and lameness examination was performed at walk and trot by an equine surgeon specialist (A.S.G.) before the experiment to ensure that all of the horses used were healthy and sound. Full details of the exercise regimen are provided below.

### 2.2. Microchipping

Percutaneous thermal sensing microchips (LifeChip^®^ with Bio-thermo^TM^; Destron Fearing^TM^; TX, USA), which contained a passive transponder programmed temperature sensor, were implanted two weeks before the commencement of the study. The horses were placed in a crush and sedated using xylazine (0.3–0.4 mg/kg body weight (BDW IV)). The implantation sites were clipped and surgically prepped using betadine and alcohol. Three milliliters of local anesthetic (Lignocaine Hydrochloride 20 mg/mL) was injected subcutaneously five minutes before implantation of the microchips. The microchip was inserted perpendicular to the skin to the maximum depth allowed by the presterilized 12-gauge needle assembly containing the transponder.

The sites were determined using set parameters to ensure uniformity of their position. The percutaneous thermal sensing microchips were implanted into the nuchal ligament (only in the first two horses) halfway between the poll and the withers into the right splenius muscle halfway between the poll and the middle of the scapular spine, into the right gluteal muscle halfway between the tail head and the right tuber coxae, and into the right pectoral muscle in the middle of the right cranial pectoral muscle. The nuchal ligament PTSM was inserted dorsally following the same guidelines insertions as the conventional ID microchips.

The position of the microchips was followed up with ultrasound (MyLab Delta; Esaote S.p.A; Genova; Italy) examinations with a linear transducer (3–11 MHz frequency) to detect any abnormalities in the surrounding area after two months of implantation.

A preliminary study using two horses, fitted with a central venous temperature probe (see below) and PTSMs placed in the nuchal ligament and right gluteal muscle area, was undertaken. After observing a weak correlation and statistically nonsignificant data (*r* = 0.01, *p* = 0.93) between the nuchal ligament temperatures and central venous temperatures, the study design was modified so that PTMS were placed in the right splenius muscle, the right pectoral muscle, and the right gluteal muscle of 8 horses (including the two used in the preliminary study). The present study includes the results of these 8 horses.

### 2.3. Central Venous Temperature (T_CV_) Probe Insertion

A small area in the cranial third of the jugular groove was clipped and aseptically prepared before the initiation of the treadmill exercise. A type T flexible implantable thermocouple (Physitemp Instrument; Clifton, NJ, USA) was introduced into the jugular vein through the lumen of a 14 Gauge 3.25 inches (8 cm) intravenous catheter (Angiocath^TM^; Becton-Dickinson and company; Franklin Lakes, NJ, USA). The thermocouple was introduced 80 cm from the IV catheter within the jugular vein toward the thorax. The temperature was displayed on a monitor (Thermalert model TH-8; Physitemp Instrument; Clifton, NJ, USA).

### 2.4. Rectal Temperature (T_R_) Probe Placement

A temperature data logger (HOBO Pro v2; U23-002; Onset Computer Corporation; Bourne, MA, USA) was used to obtain rectal temperature (T_R_) during exercise. A 184 cm long thermal sensor was fed through a universal insemination pipette for mares and introduced 40 to 50 cm into the rectum. Fecal content was removed prior to insertion of the thermal sensor. The data logger was secured to the tail using vet wrap.

### 2.5. Data Collection

#### 2.5.1. Exercise Program

All horses were familiarized and habituated to the treadmill (Veterinary Pit Model 980; Classic Treadmills Australia PTY LTD; QLD, Australia) and the microchip scanners two days before data collection. Because all horses were exposed to the treadmill in previous studies, only one session of habituation was required. This consisted of walking speed at 2 m/s for approximately 2 min followed by trotting at 4 m/s for 2.5 min and cantering at 6 to 7 m/s for 1 min. All horses moved comfortably on the treadmill and transition well between the different gaits. No signs of distress or incoordination were observed during habituation. Microchip scanners were used while the horses were exercised to habituate them to the operators’ movement and scanner noise.

During the experiment, each horse underwent a standardized treadmill exercise program, where slight variation on the speed (max speed range between 8 and 10 m/s) and duration (between 8 and 11.5 min) of the exercise was adapted to their individual fitness levels. The horses were warmed up by walking them for 5 min in an undercover area with a hard floor surface next to the treadmill room. Horses were walked in this area both pre- and postexercise. The exercise began at the speed of 2 m/s for 30 s, with the treadmill being set at a 5% incline. The speed was then increased to 4 m/s and maintained for 2.5 min. The speed was then increased from 6 to 8 m/s to change the gait from trotting to cantering and then to galloping. Once the gait changed, the speed increased by 1 m/s every minute until the T_CV_ reached 41 °C. When the T_CV_ reached 41 °C, the treadmill speed was reduced to 4 m/s and maintained at this speed for 2 min. Two wall fans were operated in front of the treadmill machine with approximately a 1.5 m distance and 2.5 m from the floor on the wall at 5 m/s just above the level of the treadmill during the exercise. Following the 10 min of hand-held walking, the horses were washed with cold tap water and returned to the small yards near the Equine Specialist Hospital.

#### 2.5.2. Temperature Acquisition

The microchip temperature in the nuchal ligament (T_NL_), the right splenius muscle (T_SM_), the right gluteal muscle (T_GM_), and the right pectoral muscle (T_PM_) were measured using microchip scanners (GPR+; Destron Fearing^TM^; Dallas, TX, USA). Two scanners were used so that the temperature of the different sites could be obtained at the same time. Microchip temperatures were acquired once before the commencement of treadmill exercise and then every 30 s during the treadmill exercise. The temperatures were then measured at one-minute intervals during the cool-down walk phase. Central venous temperature (T_CV_) and rectal temperature (T_R_) data were obtained at 1 s intervals. The T_CV_ and T_R_ data were extracted to match the time points of the microchip temperature measurements.

#### 2.5.3. Data Processing

Temperature data (T_CV_, T_R_, T_SM_, T_GM_, T_NL_, T_PM_) were pooled from the eight horses to document the changes in the body temperature during the experiment. The temperatures were grouped into the following phases: prior to exercise (Phase A; static phase), during exercise on the treadmill (Phase B; dynamic phase), immediately after exercise (Phase C; static phase), and cool-down walk (Phase D; dynamic phase) (Figure 1).

Eight temperature readings were recorded per horse for each temperature acquisition site during Phases A and C. A minimum of 15 temperature readings were recorded per horse for each temperature acquisition during Phase B, and a minimum of 8 temperature readings were recorded per horse for each temperature acquisition during Phase D (Figure 1).

### 2.6. Statistical Analysis

All of the PTSM temperatures and T_R_ were paired with T_CV_ (e.g., T_CV_/T_PM_ pair, T_CV_ /T_GM_ pair, T_CV_ /T_SM_ pair, and T_CV_/T_R_ pair) to calculate the correlation and differences between T_CV_, the core body temperature, and the other body temperatures prior to, during, and after treadmill exercise.

All statistical analyses were conducted in R [38]. The normal distribution of the data was investigated using a Shapiro–Wilk test. The significance level was set at *p* < 0.05.

The correlation coefficients of the temperature readings were computed to determine the relationships between temperature pairs using Pearson and Spearman rank tests. In addition, the limit of agreement (LOA) was computed to assess the agreement in temperature readings obtained using T_CV_, T_GM_, T_SM_, T_PM_, and T_R_. For each pair, a generalized linear regression model was fitted to get the mean bias and the standard error for computing the corresponding LOA.

For Phases B and D as dynamic phases, the repeated correlation coefficient for normally distributed data and the bootstrap method for non-normal data were used. The linear mixed-effects model was fitted to compute the LOA for the repeated temperature readings obtained using T_CV_ and PTSM.

The correlation coefficients in the current study were determined based on previously used categories (|r| = 1: Perfect correlation, 0.9 > |r| > 0.7: Strong correlation, 0.6 > |r| > 0.4: Moderate correlation, 0.3 > |r| > 0.1: Weak correlation, and r = 0: Zero correlation) [39].

For theoretical explanation and model formula used in the analysis, refer to Appendix A.

## 3. Results

The average depth of the microchip from the skin surface to the right pectoral muscle was 2.01 cm, to the right gluteal muscle it was 2.36 cm, and to the right splenius muscle, it was 2.14 cm (Figure 2).

### 3.1. Results of the Preliminary Study (n = 2)

In the preliminary study (*n* = 2), the T_CV_ had the highest correlation with T_GM_ (*r* = 0.84, *p* < 0.001). There was no statistically significant relationship between T_NL_ and T_CV_ (*r* = 0.01, *p* = 0.93) or between T_R_ and T_CV_ (*r* = 0.13, *p* = 0.35). However, the correlation between T_NL_ and T_R_ was strong (*r* = 0.80, *p* < 0.001).

### 3.2. Results of the Final Study (n = 8)

#### Body Temperatures and Its Paired Analysis

Most of the recorded body temperatures were normally distributed. Non-normal distribution was found in T_R_ (*p* = 0.012) during Phase A, in T_SM_ (*p* = 0.023) during Phase C and in T_GM_ (*p* = 0.008) during all phases (Phases A–D). Further details of these analyses are shown in Table A1 of Appendix A.

The changes in temperature at the different areas of the body are shown in Figure 3. The central venous temperature peaked at the end of Phase B, whereas the PTSMs temperatures peaked during Phase C (T_PM_ and T_GM_) and the first minute of Phase D (T_SM_). Rectal temperature increased until the end of the test, whereas the other body temperatures decreased during Phase D.

Table 1 summarizes the horse body temperature readings from each method for the various phases of the experiment.

During Phase A, the lowest temperature was 36.90 °C at T_PM_ and the highest observed was 39.10 °C at T_SM_. Statistically significant and strong correlations were observed for T_CV_/T_PM_ (*r* = 0.95, *p* < 0.001) and for T_CV_/T_SM_ (*r* = 0.87, *p* < 0.001) pairs. There were no other statistical correlations.

During Phase B, both the lowest and highest temperatures were recorded at T_SM_ (37.82 °C and 40.37 °C, respectively). Statistically significant and strong correlations between temperature readings were observed for T_CV_ /T_GM_ (*r* = 0.84, *p* = 0.01), T_CV_/T_SM_ (*r* = 0.83, *p* = 0.01), and T_CV_/T_R_ (*r* = 0.83, *p* = 0.01) pairs, but no significant correlations were observed for the T_CV_/T_PM_ pair (*p* = 0.10).

During Phase C, the lowest temperature was 38.53 °C for T_R_ and the highest was 42.00 °C at T_SM_. There was no statistically significant correlation in the temperature pairs in this phase.

During Phase D, the lowest temperature was 38.91 °C for T_PM_ and the highest was 41.61 °C at T_SM_. There was no statistically significant correlation in the temperature pairs in this phase.

Combining all the data (Phases A–D), the lowest temperature was 36.90 °C for T_PM_ and the highest was 42.00 °C for T_SM_. Statistically significant and strong correlations were observed in all the temperature pairs. The highest correlation was observed in the T_CV_/T_PM_ pair (*r* = 0.94, *p* < 0.001), and the second-highest correlation was in the T_CV_/T_GM_ pair (*r* = 0.90, *p* < 0.001). The lowest correlation was between that of T_CV_ and T_R_ (*r* = 0.71, *p* < 0.001).

### 3.3. Results of Limit of Agreement (LOA)

The average discrepancy between the methods (the bias indicated by a horizontal red line in the plots) was small (Figure 4, Figure 5, Figure 6 and Figure 7). The limits of agreement (LOA) were narrow, and it can be inferred that the methods are essentially equivalent in measuring horse body temperature at various phases of the experiment (Figure 4, Figure 5, Figure 6 and Figure 7).

During Phase A, higher temperatures were recorded for T_SM_ (bias = −0.088) and T_R_ (bias = −0.195) than T_CV_. Lower temperatures were obtained from T_GM_ (bias = 0.388) and T_PM_ (bias = 0.400) compared with T_CV_.

During Phase B, temperatures obtained using T_CV_ were higher than all the other methods.

During Phases C and D, higher temperatures were recorded at T_GM_, T_SM_, and T_PM_ compared with T_CV_.

Similarly, Table A2 (in Appendix A) presents separate LOAs for the temperature readings taken during the various phases of the experiment.

### 3.4. Results of Repeated Measured Correlation (r_mc_) during Phases B and D

The repeated-measures correlation coefficients (r_mc_) for body temperatures acquired during Phases B and D indicated that there were significant correlations for all temperature pairs. The resulting r_mc_ along with 95% bootstrap confidence intervals and *p*-values are summarized in Table 2. A strong positive correlation was calculated in the pair of T_CV_/T_PM_ (r_mc_ = 0.93, *p* < 0.001), followed by the T_CV_/T_GM_ pair (r_mc_ = 0.88, *p* < 0.001) and the T_CV_/T_SM_ pair (r_mc_ = 0.73, *p* < 0.001). Finally, a moderate positive correlation was observed in the pair of T_CV_/T_R_ (r_mc_ = 0.51, *p* < 0.001) (Table 2).

During Phases B and D, the muscle temperatures were higher than T_CV_ (T_CV_/T_GM_, bias = −0.459; T_CV_/T_SM_, bias = −0.260; and T_CV_/T_PM_, bias = −0.319), whereas the T_R_ was lower than T_CV_ (bias = 0.483). The average discrepancy between the temperatures (the bias indicated by a horizontal red line in the plots) was small (Figure 8). The limits of agreement (LOA) were narrow. Hence, with a narrow LOA and small bias, it can be inferred that the temperatures were essentially equivalent in measuring horse body temperature at various phases of the experiment. The scatter around the bias line gets larger as the average temperature readings get higher. Therefore, the consistency was decreased across the graph as the average increases (Figure 8).

Similarly, Table A3 (in Appendix A) presents separate LOAs for the temperature readings taken during Phases B and D.

## 4. Discussion

Percutaneous thermal sensing microchips have been previously used in a few horse studies [35,37]; however, this is the first study reporting the use of PTSMs in horses during and postexercise. The results from our data revealed that the optimal location for implantation of a PTSM is the pectoral muscle based on the strong correlation in the T_CV_/T_PM_ pair followed by the T_CV_/T_GM_ and T_CV_/T_SM_ pairs during strenuous exercise on the treadmill and cool-down walk immediately after the exercise.

The protocol of the preliminary study was modified due to the weak correlation between T_NL_ and T_CV_ during and immediately after exercise in the first two horses assessed. This finding is most likely related to the poor vascular supply of the nuchal ligament compared to other muscles, and this could explain the delayed temperature changes observed in T_NL_ compared to T_CV_ [40,41]. The study of Robinson et al. [35] revealed that the PTSM in the nuchal ligament was sensitive to the surrounding environmental factors, such as ambient temperature, so the study recommended another body site to implant the microchip for reading body temperature.

Differences between the body temperatures observed at the end of Phases B and D (Figure 3) were due to the normal physiologic reaction of thermoregulation [18]. During exercise, an increase in metabolic heat production augments the rate at which heat in muscles can be dissipated to the environment in order to prevent dangerous elevations in tissue temperature. Upon the cessation of exercise, the rate of heat production may greatly exceed the rate of heat dissipation, leading to a sustained elevation of muscle temperature. During short-term, high-intensity exercise, the rate of heat production will exceed the rate of heat loss throughout the exercise, and the body temperature will continue to increase until the cessation of exercise. In a study by Marlin et al. [42], similar results were reported during 21 min of exercise on the treadmill (up to 10 m/s of speed). The muscle, pulmonary artery, and rectal temperatures increased to 5.0 °C, 4.8 °C, and 1.7 °C, respectively. In the present study, similar outcomes were observed during 10 min of exercise (up to 10 m/s of speed). The PTSM temperatures in the muscles increased the most (3.2 °C), followed by central venous temperature (2.8 °C) and rectal temperature (0.98 °C). In this instance, a large proportion of the metabolic heat load will be dissipated during the postexercise period. The postexercise temperature response (rate of temperature decay) is influenced by convective heat transfer between blood and muscle and conductive heat transfer within the muscle. In the current study, PTSM temperatures followed the same pattern of a continuous increase of temperature until a few minutes after cessation of exercise and then a gradual decrease during the cooling phase. However, the muscle temperature differed depending on the location of the microchips. These differences could be due in part to factors such as differences in the muscle mass and energy turnover efficiency of the muscles used in the exercise activity and work intensity of those muscles [43,44,45].

The depth and location of the PTSM within the same muscle group may be different between horses. Different temperature gradients within the muscle both at rest and during exercise have been reported in humans [46,47,48]. Most commonly, the highest temperature is found in the deepest part of the muscle. Different parts of the same muscle may have different activity levels during exercise as well as different blood flows [46]. In humans and laboratory animals, it has been demonstrated that aging causes a profound redistribution of skeletal muscle blood flow within and between muscle groups [49]. During the whole exercise program (Phases A–D), T_GM_ had a non-normal distribution (Table A1). It may due to the various depths of microchip location in the biggest muscle or different muscle compartments work individually [46,47].

Although the different muscle sites used in the present study actively engage differently during exercise, temperature change followed a very similar pattern, with all rising rapidly to exceed rectal temperature. Previous studies found that PTSMs were a reliable alternative to rectal thermometry for the measurement of body temperature in equids at rest in an ambient temperature >15.6 °C [50]. In the current study, temperatures obtained using PTSMs had a poor correlation with rectal temperature during exercise. This is likely due to thermal inertia and a higher dependence on conductive heat transfer by blood. Rectal temperature responds more slowly in comparison to core body temperature and PTSM temperatures [51,52]. Ambient temperature may have affected PTSM values. The ambient temperature during exercise in our study was between 18.9 °C and 21.7 °C. In a previous study, the effect of ambient temperature on PTSMs placed in the nuchal ligament was evaluated in horses at rest. In that study, the PTSMs underestimated rectal temperatures <38.9 °C and overestimated rectal temperatures ≥38.9 °C at an ambient temperature of 21.2 °C [35].

Obtaining rectal temperatures in racehorses immediately after a race is both time-consuming and risky for the operator. In the current study, it was found that the mean temperature differences between T_CV_ and T_R_ in the period between 3 and 5 min in Phase D were the least among the other temperature pairs during the same period, suggesting that the attainment of rectal temperature has value postrace. However, the lower correlation between the temperatures during Phase D and the risk factors to handlers may negate the value. T_R_ showed a moderate correlation with T_CV_ during the whole treadmill test, but it was weaker than the microchip temperatures. Based on the results from the current study, it is recommended that rectal temperature should not be used as a measurement postexercise, not only for safety reasons but also because of the poor correlation with the central venous temperature immediately after exercise and at least 8 min postexercise. Since it is believed that the first 8 to 10 min after the race is the most likely period to detect horses with emerging signs of EHI, it was found that only one study collected the temperature after the cool-down phase (8 to 10 min postexercise) of the experiment [53]. In the current study, T_CV_ began to decrease as soon as exercise at maximum speed ceased in Phase B, and T_R_ continuously increased until the end of Phase D. The continuous increase in T_R_ has been observed in previous studies [23,54] during the first 10 min of walking recovery when horses were allowed to cool passively. This resulted from the heat redistribution produced by muscle activity to the skin and other compartments (gastrointestinal tract including the rectum) [55,56].

Intramuscular implantation of the PTSM is minimally invasive, requiring only the injection of the microchip through a large gauge needle like the conventional ID microchip insertion [57]. After initial implantation, measuring body temperature is completely noninvasive. No adverse reactions were observed concerning the microchip implantation site, and the procedure was well tolerated by all horses. Microchip migration is often suggested as a problem [58]; however, the migration or movement of the microchips was not observed in the short follow-up period of this experiment, and the readability of the microchips was obtained six months after implantation (unpublished data). The horses showed no limitations to movement by having the microchips placed in their muscles. Other advantages of using PTSMs include the speed and ease of use, life-long battery life of the microchips, and the rechargeable microchip reader, and it is also possible to include the unique identification number assigned to each horse [35,37].

A strong positive correlation (*r* = 0.84, *p* < 0.001) was found between gluteal percutaneous thermal sensing microchips, and central venous temperatures prompted the investigation for additional muscle sites to implant microchips. Even though the current study showed a high correlation between core body temperature and PTSM in the gluteal region, near the hind limbs, and pectoral, between the front legs, measurements were considered to be an unsafe location when handling potentially excitable horses postexercise or when showing EHI signs. Acquisition of the temperatures during exercise was possible due to the experiment design on the treadmill. However, obtaining PTSM temperatures during exercise in field conditions will have limited practical application.

The protocol within the current study was for individual horses to reach a central venous temperature of 41 °C. As unconditioned horses were used in the study and each animal presented different performance abilities, the exercise protocol varied slightly between horses, and this may have influenced the results. However, the core body temperature increased by 1 °C/min during exercise obtained in the current study is in agreement with what has been published for horses during high-speed exercise [8].

The goal of the current study was to analyze the relation of temperatures obtained from various sites with the central venous temperature. In this study, in all locations of PTSMs, strong positive correlations between the core temperature and microchip temperatures were noted prior to, during, and immediately after strenuous exercise on the treadmill, while rectal temperature had a moderate correlation with the core body temperature. Prior to the exercise, only the pectoral and splenius muscles had a statistically significant strong positive correlation with T_CV_. However, during this phase, the temperature data were obtained only once, which generates a low sample size. It is required to collect more data in order to assess the correlation between the muscle and central venous temperatures at rest. This work has focused largely on the analysis of temperatures during and immediately after exercise. All the different locations of PTSMs had strong positive correlations with the core body temperature during and immediately after exercise on the treadmill, while rectal temperature had a moderate correlation with the core body temperature. As it was inferred previously, the strong correlations between PTSMs temperature and T_CV_, as well as narrow LOA, PTSMs in the different muscle sites can estimate the core body temperature during and immediately after exercise on the treadmill. However, more work is needed to validate the data presented here under racetrack conditions. The use of PTSMs also would allow for further investigation in the most efficient cooling techniques and, ultimately, the best procedures to decrease EHI prevalence at the racetrack.

## 5. Conclusions

In summary, this study demonstrated the easy use of PTSMs for measuring the body temperature of horses during and immediately after exercise. The pectoral muscle was the most reliable implantation site for PTSM to track temperature changes among the three different muscles in this experimental study.

These results are promising in regards to finding a simple, safe, quick, accurate, and noninvasive method to measure the body temperature of horses immediately after high-speed exercise

Future studies are needed to validate this method under field conditions and in equine athletes working in extreme environments and intensive activity in various equestrian sports.

## Figures and Tables

**Figure 1 animals-10-02274-f001:**
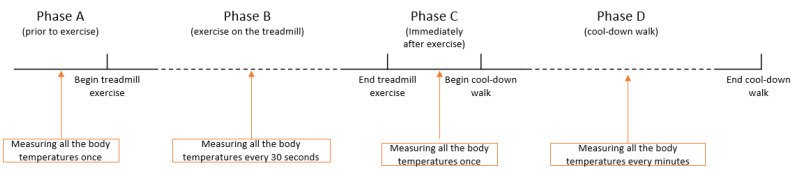
Exercise program by phases and body temperature acquisition points.

**Figure 2 animals-10-02274-f002:**
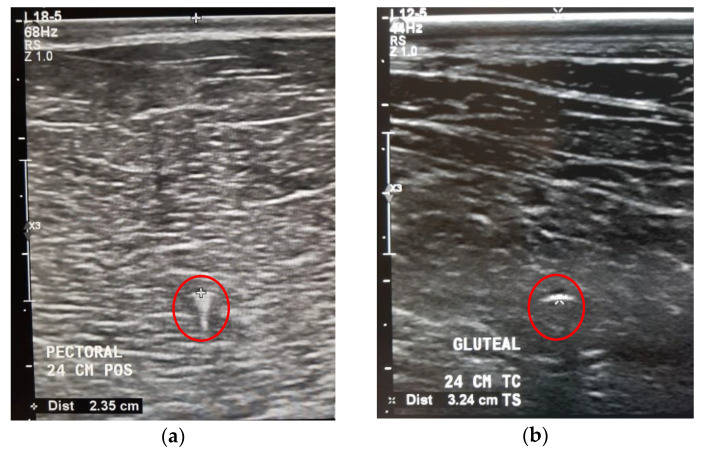
Ultrasound images of the right pectoral muscle (**a**) and right gluteal muscle (**b**) with a hyperechoic linear-structured microchip (red circle) within the muscle.

**Figure 3 animals-10-02274-f003:**
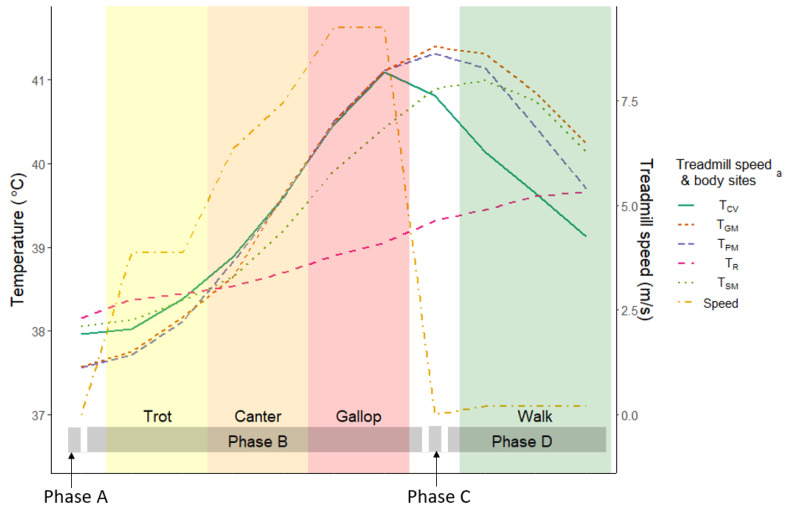
The average temperature during Phases A (prior to exercise), B (exercise on the treadmill), C (immediately after exercise), and D (cool-down walk) of the modified final study. **^a^** T_CV_ = central venous temperature, T_PM_ = pectoral muscle temperature, T_GM_ = gluteal muscle temperature, T_SM_ = splenius muscle temperature, T_R_ = rectal temperature.

**Figure 4 animals-10-02274-f004:**
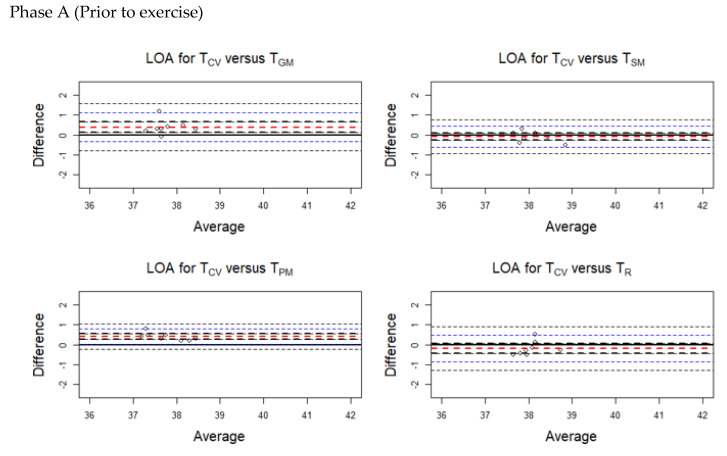
Limits of agreement (LOA) plots for the temperature pairs (T_CV_/T_GM_, T_CV_/T_SM_, T_CV_/T_PM_, and T_CV_/T_R_) during Phase A (prior to exercise). The red horizontal dashed line indicates the mean bias. The blue horizontal dashed lines indicate the upper and lower limits. The black horizontal dashed lines indicate the 95% confidence intervals of the LOA. T_CV_ = central venous temperature, T_PM_ = pectoral muscle temperature, T_GM_ = gluteal muscle temperature, T_SM_ = splenius muscle temperature, T_R_ = rectal temperature.

**Figure 5 animals-10-02274-f005:**
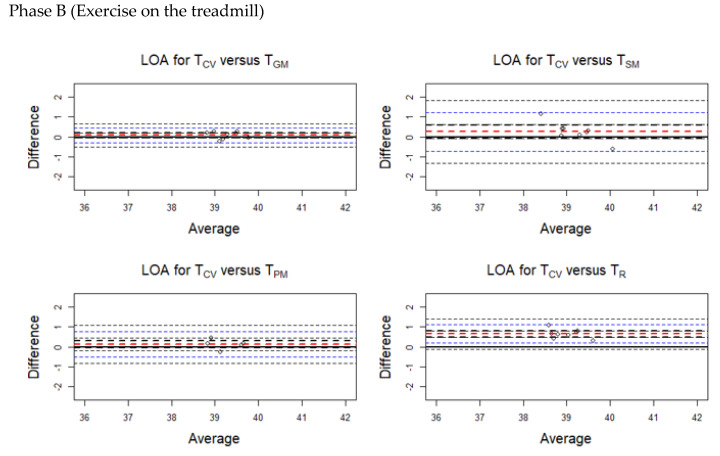
Limits of agreement (LOA) plots for the temperature pairs (T_CV_/T_GM_, T_CV_/T_SM_, T_CV_/T_PM_, and T_CV_/T_R_) during Phase B (exercise on the treadmill). The red horizontal dashed line indicates the mean bias. The blue horizontal dashed lines indicate the upper and lower limits. The black horizontal dashed lines indicate the 95% confidence intervals of the LOA. T_CV_ = central venous temperature, T_PM_ = pectoral muscle temperature, T_GM_ = gluteal muscle temperature, T_SM_ = splenius muscle temperature, T_R_ = rectal temperature.

**Figure 6 animals-10-02274-f006:**
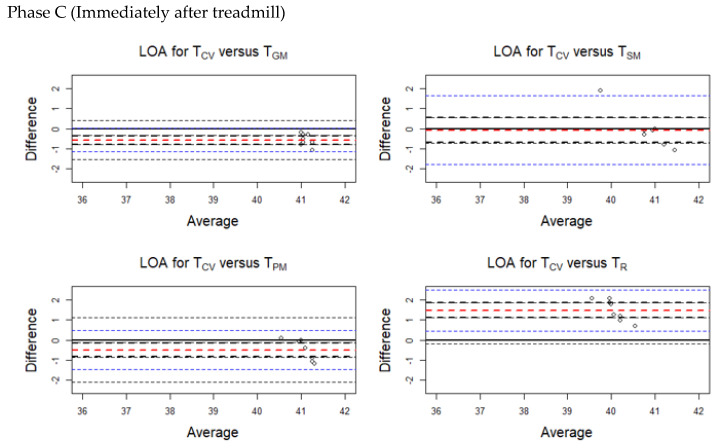
Limits of agreement (LOA) plots for the temperature pairs (T_CV_/T_GM_, T_CV_/T_SM_, T_CV_/T_PM_, and T_CV_/T_R_) during Phase C (immediately after treadmill). The red horizontal dashed line indicates the mean bias. The blue horizontal dashed lines indicate the upper and lower limits. The black horizontal dashed lines indicate the 95% confidence intervals of the LOA. T_CV_ = central venous temperature, T_PM_ = pectoral muscle temperature, T_GM_ = gluteal muscle temperature, T_SM_ = splenius muscle temperature, T_R_ = rectal temperature.

**Figure 7 animals-10-02274-f007:**
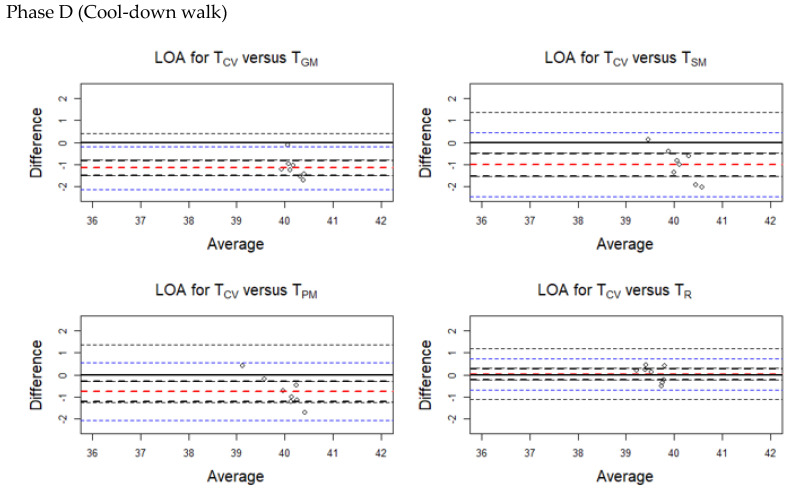
Limits of agreement (LOA) plots for the temperature pairs (T_CV_/T_GM_, T_CV_/T_SM_, T_CV_/T_PM_ and T_CV_/T_R_) during Phase D (cool-down walk). The red horizontal dashed line indicates the mean bias. The blue horizontal dashed lines indicate the upper and the lower limit. The black horizontal dashed lines indicate the 95% confidence intervals of the LOA. T_CV_ = central venous temperature, T_PM_ = pectoral muscle temperature, T_GM_ = gluteal muscle temperature, T_SM_ = splenius muscle temperature, T_R_ = rectal temperature.

**Figure 8 animals-10-02274-f008:**
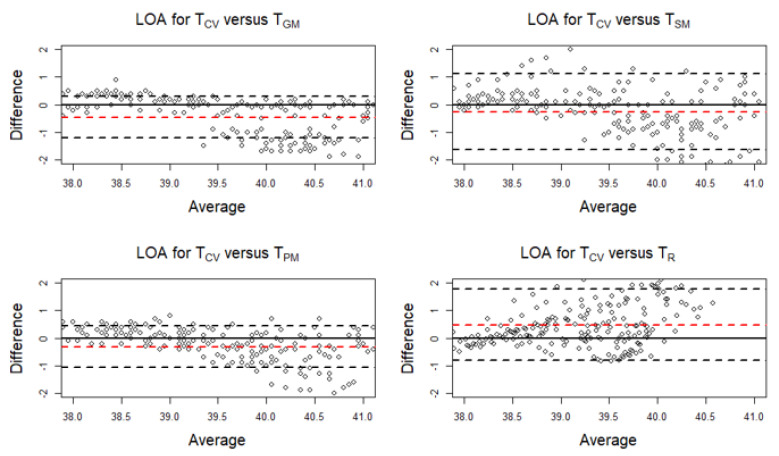
Limits of agreement (LOA) plots for the temperature pairs (T_CV_/T_GM_, T_CV_/T_SM_, T_CV_/T_PM_, and T_CV_/T_R_) during Phases B (exercise on the treadmill) and D (cool-down walk). The red horizontal dashed line indicates the mean bias. The black horizontal dashed lines indicate the upper and the lower limit. T_CV_ = central venous temperature, T_PM_ = pectoral muscle temperature, T_GM_ = gluteal muscle temperature, T_SM_ = splenius muscle temperature, T_R_ = rectal temperature.

**Table 1 animals-10-02274-t001:** Summary of body temperature readings of horses before, during, and after treadmill exercise.

	Summary Stat					Correlation (95% CI)			
	Mean	SD	Min	Max	CC	Correlation (r)	*p*-Value	LL	UL
Phase A (Prior to exercise)									
T_CV_	37.96	0.41	37.40	38.60	1.09	1	<0.001		
T_PM_	37.56	0.55	36.90	38.30	1.47	0.95	<0.001	0.76	0.99
T_GM_	37.58	0.41	37.00	38.30	1.08	0.59	0.12	−0.20	0.91
T_SM_	38.05	0.54	37.40	39.10	1.43	0.87	<0.001	0.44	0.98
T_R_	38.09	0.19	37.89	38.87	0.49	0.31	0.45	−0.50	0.83
Phase B (Exercise on the treadmill)									
T_CV_	39.31	0.32	38.91	39.75	0.82	1	<0.001		
T_PM_	39.17	0.43	38.68	39.58	1.10	0.81	0.10	−0.25	0.99
T_GM_	39.23	0.33	38.72	39.78	0.84	0.84	0.01	0.33	0.97
T_SM_	39.05	0.73	37.82	40.37	1.87	0.83	0.01	0.31	0.97
T_R_	38.66	0.42	38.06	39.46	1.10	0.83	0.01	0.30	0.97
Phase C (Immediately after exercise)									
T_CV_	40.81	0.14	40.60	41.00	0.33	1	<0.001		
T_PM_	41.31	0.48	40.50	41.90	1.15	−0.09	0.83	−0.75	0.66
T_GM_	41.39	0.22	41.10	41.80	0.54	−0.42	0.30	−0.87	0.41
T_SM_	41.00	0.28	38.80	42.00	0.67	0.62	0.10	−0.16	0.92
T_R_	39.32	0.53	38.53	40.20	1.34	0.13	0.76	−0.63	0.76
Phase D (Cool-down walk)									
T_CV_	39.59	0.20	39.31	40.00	0.50	1	<0.001		
T_PM_	40.36	0.74	38.91	41.29	1.84	0.5	0.20	−0.31	0.89
T_GM_	40.77	0.38	40.11	41.25	0.92	−0.43	0.29	−0.87	0.39
T_SM_	40.61	0.70	39.40	41.61	1.72	−0.11	0.79	−0.76	0.64
T_R_	39.56	0.35	39.12	39.99	0.88	0.21	0.62	−0.58	0.8
All the data (Phases A–D)									
T_CV_	39.42	1.07	37.40	41.00	2.70	1	<0.001		
T_PM_	39.65	1.59	36.90	41.90	4.00	0.94	<0.001	0.87	0.97
T_GM_	39.95	2.53	37.00	41.80	6.34	0.90	<0.001	0.80	0.95
T_SM_	39.65	1.37	37.40	42.00	3.46	0.82	<0.001	0.67	0.91
T_R_	38.92	0.68	37.89	40.20	1.75	0.71	<0.001	0.47	0.85

SD = standard deviation, CC = correlation coefficient, CI = confidence interval, LL = lower limit, UL = upper limit, T_CV_ = central venous temperature, T_PM_ = pectoral muscle temperature, T_GM_ = gluteal muscle temperature, T_SM_ = splenius muscle temperature, T_R_ = rectal temperature.

**Table 2 animals-10-02274-t002:** Repeated-measures correlation coefficient (r_mc_) for body temperature readings in horses during Phases B (exercise on the treadmill) and D (cool-down walk).

r_mc_		*p*-Value	Lower Limit	Upper Limit
T_PM_	0.93	<0.001	0.88	0.93
T_GM_	0.88	<0.001	0.84	0.91
T_SM_	0.73	<0.001	0.62	0.79
T_R_	0.51	<0.001	0.43	0.62

T_CV_ = central venous temperature, T_PM_ = pectoral muscle temperature, T_GM_ = gluteal muscle temperature, T_SM_ = splenius muscle temperature, T_R_ = rectal temperature.

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
