# Peer review of "The Use of Percutaneous Thermal Sensing Microchips for Body Temperature Measurements in Horses Prior to, during and after Treadmill Exercise"

_animals, 2020, doi:10.3390/ani10122274_

Round 1

Reviewer 1 Report

Dear authors,

You did a lot of work on improving the article and I appreciate forgoing the thermographic results and the quality of statistical analysis. Now, in my opinion, the quality of your manuscript is high and it is worth being present for the wide community. Please see some minor recommendations, corrections, and questions formulated during the revision process of your manuscript and use them as considered to increase the value of your paper.

Abstract

A minor correction of the abstract structure is required. Please, see the abstract guidelines. The structure of the abstract should include (without headings) (1) Background; (2) Methods; (3) Results; and (4) Conclusions. You have placed the question addressed in a broad context and highlight the purpose of the study, maybe a little bit too long, but correctly. Then, you have briefly described the main methods or treatments applied, and this part is really good. However, the summarize the article's main findings are poor (without the main results) and the main conclusions or interpretations are not at all indicated. Please, rewrite the abstract according to the general guidelines of Animals journal.

L 59-60 ... unlike other animal athletes, such as greyhounds or humans, ... Please, rephrase this sentence. It seems like humans will be animal athletes.

L 73 Please, delete "However" and explain "some conditions"

L 76-77 One short sentence is missing, linking the core body temperature measurement and hyperthermia into a logical sequence

L 82 remove space before dot

L 84 an abbreviation WBGT is extended in line 81. It is not necessary explain it again here

L 85-86 remove spaces after "Racing Queensland", "Australia", "Wales" and "Victoria"

L 90 remove space before dot, at the beginning of this line

L 91 remove space before "a horse with EHI"

L 95 remove space before dot, at the beginning of this line               

L 96 remove space after "coma" and before dot

L 108 Please, add "however" at the beginning of following sentence "Rectal thermometry is the most commonly"

L 112 remove space before dot

L 117 add a dot after "the rectal temperature"

L 119 Please, add references

L 123 and L 413 maybe the word "relation" is more suitable than "correlation" here as a more general (in your results both correlations and their lack were described)

L 140 Please, briefly describe which kind of examination did you perform

L 147 remove one additional bracket

L 159 Please, add a model of the ultrasound and the type and frequency of the probe

L 163 only weak correlations or weak and statistically insignificant?

L 184 remove space after "the microchip scanners"

L 185 Please, add references of your previous studies on the treadmill if they were published

L 190 remove space after "movement and scanner noise"

L 197 Please, rephrase "increased to 6 to 8 m/s"

L 252 add zero after r=0.8. It is recommended to use always the same number of decimal places (also apply to the temperature values in the lines 279-280)

Please, use always the same style of p value. In lines L 250-250 is p, whereas in lines 257-260 is p.

L 288 Figures not Figure

Please, edit the figures 3, 4, 5, 6, 7, and 8 descriptions as continuous, justified text with enter

L 310 remove two spaces in the bracket with "rmc = 0.73, p<0.001"

In Table 1 as in whole manuscript body, the p-value is described as "Significance (p)", whereas in Tables 2, 3 and 4 as p-value. The unification is required.

L 342 remove space before dot

L 342-344 Consider comparing your results with other post-exercise temperature measurements, also using another measurement techniques for example thermography that you withdrew, in horses. Short-term, high-intensity exercise is typical for the race horses so please, pay attention to the latest publications on temperature changes in race horses since the aim of the discussion it to compare your results with others. The single reference after the whole paragraph is not enough.

L 357 add reference

L 365 introduce here the value of your ambient temperature an briefly state if it could affected your results or not

L 399 "The high correlation" the strongest positive correlations?

L 414 "All the muscles in this study had strong correlations with the core" I suggest rephrase to "In this study, in all locations of PTSMs strong positive correlations between the core temperature and microchip temperatures were noted"

L 415 "during and immediately after" whereas in the title is also "prior". What was hapened with "a prior to"

L 422-423 still "The pectoral muscle" in not an "ideal implantation site"

Please, consider rephrase and partially replace the last paragraph of discussion and results sections, since the conclusion section should provide readers with a brief summary of the main conclusions based on your results. The better place for all speculations is the last paragraph of discussion section.

L 436 shouldn't you add the grant number?

is it intentional to mark some values in yellow in table 3 in appendices? if so, add this information in the table description

Please, explain all the abbreviations placed in tables 3,4 and 5 under them. All tables should be readable as a separated part of manuscript  

If it is possible, please a list of abbreviations, because your work has a lot and I believe that it will make the read more affordable.

Author Response

Dear authors,

You did a lot of work on improving the article and I appreciate forgoing the thermographic results and the quality of statistical analysis. Now, in my opinion, the quality of your manuscript is high and it is worth being present for the wide community. Please see some minor recommendations, corrections, and questions formulated during the revision process of your manuscript and use them as considered to increase the value of your paper.

Abstract

A minor correction of the abstract structure is required. Please, see the abstract guidelines. The structure of the abstract should include (without headings) (1) Background; (2) Methods; (3) Results; and (4) Conclusions. You have placed the question addressed in a broad context and highlight the purpose of the study, maybe a little bit too long, but correctly. Then, you have briefly described the main methods or treatments applied, and this part is really good. However, the summarize the article's main findings are poor (without the main results) and the main conclusions or interpretations are not at all indicated. Please, rewrite the abstract according to the general guidelines of Animals journal.

 Thank you for this suggestion. We have rewritten the abstract.

Accurately measuring body temperature in horses, will improve the management of horses suffering from, or being at risk of developing post-race exertional heat illness. PTSM has the potential for measuring body temperature accurately, safely, rapidly, and non-invasively. This study was undertaken to investigate the relation between the core body temperature and PTSM temperatures prior, during and immediately after exercise. The microchips were implanted into the nuchal ligament, the right splenius, gluteal and pectoral muscles and these locations were then compared with the central venous temperature, which is considered to be the “gold standard” for assessing core body temperature. The changes in temperature of each implant in the horses were evaluated in each phase (prior, during and immediately post-exercise) and combining all the phases. There were strong positive correlations ranging from 0.82 to 0.94 (p<0.001) of all the muscles sites with the central venous temperature when combining all the phases. Additionally, during the whole period, PTSM had narrow limits of agreement (LOA) with central venous temperature, which inferred that PTSM is essentially equivalent in measuring horse body temperature. Overall, the pectoral PTSM provided valid estimation of the core body temperature. (Line 40-54)

L 59-60 ... unlike other animal athletes, such as greyhounds or humans, ... Please, rephrase this sentence. It seems like humans will be animal athletes.

Thank you for this opinion. We have rephrased the sentence from “unlike other animal athletes, such as greyhounds or humans,” to “unlike humans or other animal athletes,” (Line 74-75).

L 73 Please, delete "However" and explain "some conditions"

Thank you for the comment. We have rewritten the sentence from ‘However, under some conditions,’ to ‘Under hot or hot and humid conditions,’. (Line 88-89)

L 76-77 One short sentence is missing, linking the core body temperature measurement and hyperthermia into a logical sequence

Thank you for this advice. We have rewritten the sentence from ‘When the central blood temperature reaches 42.5℃, the hypothalamic temperature reaches 41.5℃ and muscle temperatures reach 45℃,’ to ‘When the central blood, the hypothalamic, and muscle temperatures reach 42.5℃, 41.5℃, and 45℃, respectively,’. (Line 90-92)

L 82 remove space before dot

Thank you for the comment. We have deleted the space. (Line 97)

L 84 an abbreviation WBGT is extended in line 81. It is not necessary explain it again here

Thank you for the comment. We have rewritten the sentence from ‘When the forecast wet bulb globe temperature (WBGT) is..’ to ‘When the forecast WBGT is..’. (Line 99)

L 85-86 remove spaces after "Racing Queensland", "Australia", "Wales" and "Victoria"

Appreciate for this comment. We have removed the spaced. (Line 100-101)

L 90 remove space before dot, at the beginning of this line

Many thank you for this comment. We have removed the space. (Line 104)

L 91 remove space before "a horse with EHI"

Thank you for this comment. We have removed the space. (Line 106)

L 95 remove space before dot, at the beginning of this line        

Many thank you for the comment. We removed the space. (Line 110)

L 96 remove space after "coma" and before dot

Thank you for the comment. We have removed the space. (Line 111)

L 108 Please, add "however" at the beginning of following sentence "Rectal thermometry is the most commonly"

Appreciate for this opinion. We have rewritten the sentence from ‘Rectal thermometry ..’ to ‘However, rectal thermometry..’. (Line 123)

L 112 remove space before dot

Thank you for the comment. We have removed the dot. (Line 127)

L 117 add a dot after "the rectal temperature"

Many thank you for this comment. We have added a dot in the sentence. (Line 132)

L 119 Please, add references

Thank you for this comment. We have added a reference in the sentence. (Line 134)

L 123 and L 413 maybe the word "relation" is more suitable than "correlation" here as a more general (in your results both correlations and their lack were described)

Appreciate for the suggestion. We have now used ‘relation’ instead of ‘correlation’ in the sentences. (Line 138, 441)

L 140 Please, briefly describe which kind of examination did you perform

Many thank you for the comment. We have rephrased the sentence from ‘… and lameness examination were performed by an equine surgeon specialist before..’ to ‘… and lameness examination were performed at walk and trot by an equine surgeon specialist (A.S.G.) before..’. (Line 157-158)

L 147 remove one additional bracket

Thank you for the comment. We have removed the additional bracket. (Line 164)

L 159 Please, add a model of the ultrasound and the type and frequency of the probe

Thank you for the advice. We have added the information about ultrasound and probe in the sentence. From ‘with ultrasound examinations to detect’ to ‘with ultrasound (MyLab Delta; Esaote S.p.A; Genova; Italy) examinations with a linear transducer (3-11 MHz frequency) (Line 176-177)

L 163 only weak correlations or weak and statistically insignificant?

Many thank you for this comment. It was statistically insignificant, so we have added “and statistically insignificant (r=0.01, p=0.93)” in the sentence. (Line 181)

L 184 remove space after "the microchip scanners"

Thank you for this comment. We have removed the space. (Line 203)

L 185 Please, add references of your previous studies on the treadmill if they were published

Thank you for this suggestion. However, we do not have any published data.

L 190 remove space after "movement and scanner noise"

Thank you for the comment. We have removed the space. (Line 190)

L 197 Please, rephrase "increased to 6 to 8 m/s"

Many thank you for this advice. We have rephrased that sentence from “.. then increased to 6 to 8 m/s” to “.. then increased from 6 to 8 m/s”. (Line 216)

L 252 add zero after r=0.8. It is recommended to use always the same number of decimal places (also apply to the temperature values in the lines 279-280)

Thank you for your help. We have corrected the decimals.

Please, use always the same style of p value. In lines L 250-250 is p, whereas in lines 257-260 is p.

Thank you for this comment. We have changed “p” to “p”. (Line 277)

L 288 Figures not Figure

Many thank you for this comment. We have corrected the word. (Line 306)

Please, edit the figures 3, 4, 5, 6, 7, and 8 descriptions as continuous, justified text with enter

Thank you for your advice. We have undo the enters between sentences in the descriptions in the figures 3, 4, 5, 6, 7, and 8.

L 310 remove two spaces in the bracket with "rmc = 0.73, p<0.001"

Thank you for this comment. We have removed the two spaced in the sentence. (Line 328)

In Table 1 as in whole manuscript body, the p-value is described as "Significance (p)", whereas in Tables 2, 3 and 4 as p-value. The unification is required.

Many thank you for this advice. We have used “p-value” instead “Significance (p)” in Table 1.

L 342 remove space before dot

 Thank you for this comment. We have removed the space in the sentence. (Line 360)

L 342-344 Consider comparing your results with other post-exercise temperature measurements, also using another measurement techniques for example thermography that you withdrew, in horses. Short-term, high-intensity exercise is typical for the race horses so please, pay attention to the latest publications on temperature changes in race horses since the aim of the discussion it to compare your results with others. The single reference after the whole paragraph is not enough.

Thank you for this suggestion. We have added a paragraph In a study of Marlin, et al. [44], similar results were reported during 21 minutes of exercise on the treadmill (up to 10m/s of speed). The muscle, pulmonary artery, and rectal temperatures increased 5.0℃, 4.8℃, and 1.7℃, respectively. In the present study, similar outcomes were observed during 10 minutes of exercise (up to 10m/s of speed). The PTSM temperatures in the muscles increased the most (3.2℃), followed by central venous temperature (2.8℃) and rectal temperature (0.98℃).. (Line 362-367)

L 357 add reference

Thank you for this comment. We have added a reference for the sentence. (Line 380)

L 365 introduce here the value of your ambient temperature an briefly state if it could affected your results or not

Many thank you for your advice. We have added one sentence ‘, Ambient temperature may have affected PTSMs values. The ambient temperature during exercise in our study was between 18.9℃ and 21.7℃. In a previous study, the effect of ambient temperature on PTSMs placed in the nuchal ligament was evaluated in horses at rest. In that study, the PTSMs underestimated rectal temperatures <38.9℃ and overestimated rectal temperatures ³38.9℃ at an ambient temperature of 21.1℃. (Line 392-396)

L 399 "The high correlation" the strongest positive correlations?

Thank you for this comment. We have add more information about the correlation in the sentence. From ‘The high correlation found..’ to ‘Strong positive correlation (r=0.84, p<0.001) was found..’ (Line 426)

L 414 "All the muscles in this study had strong correlations with the core" I suggest rephrase to "In this study, in all locations of PTSMs strong positive correlations between the core temperature and microchip temperatures were noted"

Many thank you for this suggestion. We have rephrased the sentence from ‘All the muscles in the study had strong correlation with the core body temperature during and..’ to ‘In this study, in all locations of PTSMs strong positive correlations between the core temperature and microchip temperatures were noted prior, during and..’. (Line 442-444)

L 415 "during and immediately after" whereas in the title is also "prior". What was hapened with "a prior to"

Thank you for the comment. We have added a couple of sentences ‘Prior to the exercise, only the pectoral and splenius muscles had statistically significant strong positive correlation with TCV. However, during this phase, the temperature data collected only once prior to the exercise, which is low sample size to justify. It is required more data and study to achieve accurate results the correlation between the muscles and central venous temperatures at rest’. (Line 446-450)

L 422-423 still "The pectoral muscle" in not an "ideal implantation site"

Thank you for your opinion. We have deleted ‘ideal’ and added ‘most reliable’ instead. (Line 427-428)

Please, consider rephrase and partially replace the last paragraph of discussion and results sections, since the conclusion section should provide readers with a brief summary of the main conclusions based on your results. The better place for all speculations is the last paragraph of discussion section.

Thank you for this suggestion. We have rewritten the last paragraph on the discussion section. The paragraph is ‘The goal of the current study was to analyze the relation of temperatures obtained from various sites with the central venous temperature. In this study, in all locations of PTSMs strong positive correlations between the core temperature and microchip temperatures were noted prior, during and immediate after strenuous exercise on the treadmill, while rectal temperature had a moderate correlation with the core body temperature. Prior to the exercise, only the pectoral and splenius muscles had statistically significant strong positive correlation with TCV. However, during this phase, the temperature data was obtained only once, which generates a low sample size. It is required to collect more data in order to assess the correlation between the muscle and central venous temperatures at rest. This work has focused largely on the analysis of temperatures during and immediately after exercise. All the different locations of PTSMs had strong positive correlations with the core body temperature during and immediately after exercise on the treadmill, while rectal temperature had a moderate correlation with the core body temperature. As it was inferred previously, the strong correlations between PTSMs temperature and TCV, as well as narrow LOA, PTSMs in the different muscle sites can estimate the core body temperature during and immediately after exercise on the treadmill. However, more work is needed to validate the data presented here under racetrack conditions. The use of PTSMs also would allow for further investigation in the most efficient cooling techniques and ultimately, the best procedures to decrease EHI prevalence at the racetrack.

’ (Line 446-459)

L 436 shouldn't you add the grant number?

It is continuous support from a foundation. Therefore, there is no grant number.

is it intentional to mark some values in yellow in table 3 in appendices? if so, add this information in the table description

Many thank you for the comment. We have erased the yellow highlights in table 3.

Please, explain all the abbreviations placed in tables 3,4 and 5 under them. All tables should be readable as a separated part of manuscript  

Appreciate you for the advice. We have added all the abbreviations in the tables 3, 4 and 5.

If it is possible, please a list of abbreviations, because your work has a lot and I believe that it will make the read more affordable.

Based on the journal policy no list of abbreviations can be added.

Reviewer 2 Report

The paper has been greatly improved. There are numerous minor errors throughout, such as comma splices, added spaces, etc. These should be carefully corrected. 

Line 84: Don't need to redefine WBGT 

Line 133: Explain what is meant by "when required" 

Line 137: Include the rate at which hay was provided (% BW)

Line 391: Change to "...the procedure was well tolerated by all horses."

Line 401: Remove "in"

Line 403: "...was considered to be an unsafe location when handling..."

Line 428: Change to "sports"

Author Response

The paper has been greatly improved. There are numerous minor errors throughout, such as comma splices, added spaces, etc. These should be carefully corrected. 

Line 84: Don't need to redefine WBGT 

Thank you for this comment. We have rewritten the sentence from ‘When the forecast wet bulb globe temperature (WBGT) is’ to ‘When the forecast WBGT is’. (Line 99)

Line 133: Explain what is meant by "when required" 

Many thank you for this advice. We have rephrased the sentence from ‘Lucerne and grass hay were provided when required’ to ‘Ad-libitum access to pasture and are provided with oat hay (1.5% of body weight) once or twice a week’. (Line 148-150)

Line 137: Include the rate at which hay was provided (% BW)

Appreciate you for this suggestion. We have rewritten the sentence from ‘grass hay was provided three times daily’ to ‘1.5% of body weight of lucerne hay was provided per day’. (Line 153)

Line 391: Change to "...the procedure was well tolerated by all horses."

Many thank you for this suggestion. We have rephrased the sentence from ‘all the horses tolerated well the procedure’ to ‘the procedure was well tolerated by all horses’. (Line 418-419)

Line 401: Remove "in"

Thank you for this comment. We have removed “in” in the sentence. (Line 428)

Line 403: "...was considered to be an unsafe location when handling..."

Thank you for the suggestion. We have rephrased the sentence. (Line 430-431)

Line 428: Change to "sports"

 Many thank you for this comment. We have added ‘s’ in the sentence ‘… in various equestrian sport(s)’. (Line 468)

This manuscript is a resubmission of an earlier submission. The following is a list of the peer review reports and author responses from that submission.

Round 1

Reviewer 1 Report

An application of percutaneous thermal sensing microchips (PTMS) for body temperature measurements in horses concerning work on a treadmill is interesting and, as the authors pointed out, novel. However, before the detail revision some major points need to be clarified.

The main benefits of this work are the findings that the thermal sensing microchips can detect and monitor the horse internal body temperature during and after exercise. The aim should be focused on this findings and the interesting correlations with gold standard measurements (Tcv).

L 26, 51, and 380 How the measuring of the horse body temperature improve the welfare of horses? It is not a point of the study since the welfare indicators were not included. However this point of view is justified in the Introduction section, it should be removed from the Abstract and Conclusion sections.

L 74 when the exertional heat illness (EHI) is raised up, the major clinical signs, general pathogenesis, and the consequences for the horse organism should be clarified. EHI is repeated in the Introduction and Discussion sections as a major ceases of application of PTMS in the equine industry. It is a good direction for consideration and further research, however, requires a slightly more detailed approach at this stage also. Especially in L 94, the clinical signs of the central nervous system (CNS) are mentioned. Not all readers will be able to easily connect EHI with CNS disorders, therefore an additional commentary is indicated. Also, subclinical EHI in L 96-97 requires a comment. I agree with your opinion that if body temperature could be measure accurately, safely, quickly and non-invasively, then early detection and management of EHI would be possible. However, in L 359-360 is stated that intramuscular implantation of the PTMS is minimally but invasive. In L 103-104 the rectal temperature measurement is also mentioned as minimally invasive. It seems to me that a procedure involving sedation, local anesthesia, implantation, and skin suturing should be considered more invasive than placing a thermometer in the rectum. Please, clarify both issues.

L 113 As above I recommend the revision of the general aim of the study.

L 122 What about the housing, management, and condition of horses? Were they similar or comparable? What about the daily use of horses? Whether they were in training (if yes, what type and intensity) or standing in the stable or were in the pasture before and during the investigation. All these factors will affect the test results. More details are required.

L 134 The horse's nuchal ligament is a large elastic structure in the dorsal neck region. It supports the head and the neck movements and the continuity of the structure require to work properly. Are you sure, the intra-ligament implantation of PTMS does not affect the ligament structure? Maybe PTMS was implanted in surrounding soft tissues. If so, please add it and name it consistently in the nuchal ligament area.

L 141-142 If these two horses were then used again with PTMS placed in pectoral muscle, gluteal muscle, and splenius muscle? Please, clarify.

L 158, 160, 196 Add producer and country of origin.

L 170 The 10 m distance between the left side of the body and thermal camera is too high to obtain reliable measurements of the horse's surface body temperature. Please, see the work of Westermann et al. (2013) discussing the distance on measurement and reproducibility of thermographically determined temperatures. I realize that this distance is necessary to take a single image covering the whole horse, however, with the distance the measurement reliability decreases. It may be a reason why in the current study, all of the infrared surface temperatures obtained had a weak or no correlation with TCV (L 329-330). Please, discuss it. The emissivity of the infrared radiation camera and a temperature range is also required.

L 181 Please, add a short comment on how the horses were habituated to the treadmill. Which aspects of the habituation process were considered and when the horses were considered habituated.

L 181 Add a model of treadmill.

L 186 The exercise began with the treadmill on the speed set at 4m/s for 5 minutes. If the work on a treadmill was started with trot without initially walk? Please, check the speed during the subsequent gaits.

L 190 Did the horses leave the treadmill immediately after working on high speed without rest in the walk?

L 191 Did the horses return to the stable immediately after 8 to 10 minutes walk in hand?

L 206 Was the obtained data examined for normal distribution? Please add the details. In the Statistical analysis section, there is no information on data grouping taking into account the sampling frequency of individual measurements. Note that, depending on the different sampling rates, different data series vary in subsequent groups. The abundance of groups should be taken into account during statistical analysis.

L 242 The correlations TCV/TNL and TCV/TR were not significant, therefore cannot be considered as weak.

L 247 It is the only place where the bodyweight and climate conditions are mentioned. Please, add the necessary information to the Materials and Methods and the Discussion sections.

On Figures 2 and 3 the border points (time or value of some feature) are introduced: E BEGIN, T BEGIN, T MIDDLE, C BEGIN, C MIDDLE, M BEGIN, E END, W BEGIN, W MIDDLE, W END. Please, add the criteria of the border points determination to the Materials and Methods section also taking into account the data grouping method and the sampling frequency mentioned above.

L 255 In Body temperatures and its paired analysis section, the data of the temperature changes were presented separately for Phase A+B (entire treadmill test) and Phase B (cool-down). What about Phase A (exercise on the treadmill)?

L 269 r=0.72~0.98 means r Є {0.72; 0.98}? (similarly L 285, L 286, L 290)

L 270 Two values of r are needed: TPM (r=0.69) and TGM (r=0.46) had a moderate correlation with TCV.

L 270-271 The correlation between rectal temperature TCV (r=-0.05) is not significant (p=0.69). You cannot say that it is no correlation.

L 270-271 What about the weak significant correlation between TCV/TSM?

L 289-291 You cannot say the measured surface temperatures had a weak or no correlation when p>0.05. You demonstrated only one weak significant correlation between TCV/STGM.

The discussion should be rewritten according to all changes in the Introduction, Material, and methods and Results sections. Considering the inaccuracies indicated above, I cannot review the discussion except for three important points:

L 334 You didn't present the features of linear or non-linear correlation in the Results section.

L 347 That's not true that the locations of the microchip in the current study affected muscle temperature. The muscle temperature is affecting by effort and metabolic changes. You only can say that the muscle temperature differed depending on the location of the microchip. This is something completely different.

L 365 What is the battery life in PTMS? Is it inductively charged from the microchip scanner?

The conclusion did not correspond with the aim and the results. If the aims and results are changed, the conclusions should also be modified. You did not investigate the safety and time-consuming body temperature measurements and implantation of PTMS is not non-invasive. Did you recognize the investigated exercises on the treadmill as strenuous? Please, add the maximal speeds of each horse to the Results section. You also did not investigate post-racing EHI and welfare features of the racehorse. These are just speculations. The conclusions should be based on the results obtained.

Reviewer 2 Report

This is an interesting paper that presents information that advances the scientific literature. It is well written and clearly presented. I only have some minor concerns that are detailed below. 

Line 97: should be "measured" 

Line 108: should read as "while IT is recovering"

Line 117: remove first apostrophe 

Line 128: detail what the clinical examination included

Line 146: explain what parameters were used for placement

Line 181: explain how horses were habituated to treadmill exercise (length of sessions, speeds, if sedatives were used, etc)

Line 185: please give ranges to explain the "slight variation" 

Line 331: "...was used as a measure OF core..."

Line 351: missing period after 41'C

Line 355: more information is needed to justify the use of thermal sensing microchips in the neck area. Why is GM not mentioned here? 

Lines 413, 494: Citations have title twice

Round 2

Reviewer 1 Report

I found you have revised your manuscript following my and another reviewer’s comments.

However, there are still some problems with the statistical analysis. You didn't explain if obtained data were examined for normal distribution. Because you presented the correlations between different temperature measurement the data distribution is crucial to apply the adequate correlation test. You also didn't add the border points (time or value of some feature for example speed) for E BEGIN, T BEGIN, T MIDDLE, C BEGIN, C MIDDLE, M BEGIN, M END, W BEGIN, W MIDDLE, W END. This is the basis for data classification and should be well described. It is important to know when you decided to classified data as T BEGIN or T MIDDLE, C BEGIN or C MIDDLE, M BEGIN or M END, and W BEGIN, W MIDDLE or W END. What is the border between C MIDDLE and M BEGIN? On figure 2 and 3 only the temperature values were presented for each period however not the border points.

The minor problems were still there.

L 231 You tested PTSMs temperatures obtained by the implant in various locations of the horses' body. However, you test only three locations: TPM, pectoral muscle temperature; TGM, gluteal muscle temperature; TSM, splenius muscle temperature. Therefore you can not say that one is ideal for the whole horses' body. You are able to say which one is the best in this experimental structure. Please, verify this point of view also in the abstract, introduction (especially the aim of the study), results, discussion, and conclusion sections.

L 276 3.2.1. If, as you mentioned in response, the Bodyweight and climate conditions are not adding any new information from what it has been reported before and it was not the main aim of the study, you should consider removing the entire paragraph and related content.

L 370 There is no need to repeat the results in the discussion section. Remove "(r=0.56)".

L 387 In Westermann et al. (2013) [69] the minimal value for wind velocity below which no detectable decrease of temperature was described. Please see Westermann et al. (2013) Effects of Infrared Camera Angle and Distance on Measurement and Reproducibility of Thermographically Determined Temperatures of the Distolateral Aspects of the Forelimbs in Horses. J Am Vet Med Assoc. 2013 Feb 1;242(3):388-95. doi: 10.2460/javma.242.3.388.

L 388 The second reason could be connected with the emissivity of the thermal camera and temperature range used in this study. Please find the research describing superficial temperatures in the scapula and hindquarters regions, before and after an effort, and discuss why you use lower emissivity than it is considered reliable in horses.

L 405-407 One of the goals is raised here without any comments. Whether the goal was achieved if so based on what results. The answer is partly on L 327-330. Both parts should be combined in a logical whole. In general, the discussion section is difficult to read. It seem like a combination of loose thoughts, rather than a coherent whole. Improving the readability and transparency of discussions would also be appreciated.
